# Regional Variation in National Healthcare Expenditure and Health System Performance in Central Cities and Suburbs in Japan

**DOI:** 10.3390/healthcare10060968

**Published:** 2022-05-24

**Authors:** Yuna Seo, Takaharu Takikawa

**Affiliations:** Department of Industrial Administration, Faculty of Science and Technology, Tokyo University of Science, Noda 278-8510, Japan; 7422529@ed.tus.ac.jp

**Keywords:** regional variation, national healthcare expenditure (NHE), health system performance, Japan

## Abstract

The increasing national healthcare expenditure (NHE) with the aging rate is a significant social problem in Japan, and efficient distribution and use of NHE is an urgent issue. It is assumed that comparisons in subregions would be important to explore the regional variation in NHE and health system performance in targeted municipalities of the metropolitan area of Tokyo (central cities) and the neighboring municipalities of Chiba Prefecture (suburbs). This study aimed to clarify the differences of the socioeconomic factors affecting NHE and the health system performances between subregions. A multiple regression analysis was performed to extract the factors affecting the total medical expenses of NHE (Total), comprising the medical expenses of inpatients (MEI), medical expenses of outpatients (MEO), and consultation rates of inpatients (CRI) and outpatients (CRO). Using the stepwise method, dependent variables were selected from three categories: health service, socioeconomic, and lifestyle. Then, health system performance analysis was performed, and the differences between regions were clarified using the Mann–Whitney *U* test. The test was applied to 18 indicators, classified into five dimensions referred to in the OECD indicators: health status, risk factors for health, access to care, quality of care, and health system capacity and resources. In the central cities, the number of persons per household was the primary factor affecting Total, MEI, MEO, and CRO, and the number of persons per household and the percentage of the entirely unemployed persons primarily affected CRI. In the suburbs, the ratio of the population aged 65–74 and the number of hospital beds were significantly positively related to Total, MEI, and CRI, but the number of workers employed in primary industries was negatively related to Total and MEI. The ratio of the population aged 65–74 was significantly positively related to MEO and CRO. Regarding health system performance, while risk factors for health was high in the central cities, the others, including access to care, quality of care, and health system capacity and resources, were superior in the suburbs, suggesting that the health system might be well developed to compensate for the risks. In the suburbs, while risk factors for health were lower than those in the central cities, access to care, quality of care, and health system capacity and resources were also lower, suggesting that the healthcare system might be poorer. These results indicate a need to prioritize mitigating healthcare disparities in the central cities and promoting the health of the elderly in the suburbs by expanding the suburbs’ healthcare systems and resources. This study clarified that the determinants of NHE and health system performance are drastically varied among subregional levels and suggested the importance of precise regional moderation of the healthcare system.

## 1. Introduction

National healthcare expenditure (NHE), an estimate of the cost of treating injuries and diseases covered by insurance in medical institutions during the relevant fiscal year, has increased along with the aging rate [1,2], and it is now a major social problem in Japan. NHE includes medical expenses for inpatients (MEI), outpatients (MEO), and dental treatments (MED); pharmacy dispensing medical expenditures and food and living care expenditures during admission; and home-visit nursing medical expenditures [3]. MEI and MEO account for about 70% of NHE [4]. The elderly population is increasing rapidly, and the working generation’s population is decreasing rapidly. In this way, the public medical insurance system’s foundation has become extremely fragile. Insurance premiums cover about 50% of NHE, and about 40% is covered by public funds [4]. Looking at the share of insurance premiums rising, the burden of the people has increased annually [5,6].

Although Japan is one of the most successful countries in the world based on many health indicators, concerns are growing about increasing regional variation. With the country’s transition to a superaging society, it is indispensable to control the growth of medical expenses to a level that people can bear (i.e., moderating NHE) and, as part of this, correct the large regional disparity in medical expenses per capita, which are discussed in Japanese government [7,8]. Regional variation could have great influence on the performance of regional health systems. In Japan, variation in health outcomes, its causes, performances of regional health systems, and the effects of socioeconomic determinants of health are becoming increasingly complex with the country’s transition to a superaging society [9,10,11,12]. It is deemed essential to clarify the determinant driving regional variation of NHE and further find effective strategies to moderate it and the quality of health service. Nomura et al. (2017) insisted that progress has slowed down and health variations between prefectures are growing, although it has been successful overall in reducing mortality and disability from most major disease [13]. Tokita et al. (2000) examined the determinants of health expenditure variation at the prefecture level and stated that the numbers of hospital beds and doctors per capita are significant determinants of NHE [14]. This inequitable geographic distribution of health resources could cause regional variation in health outcomes [15,16]. In addition to supply-side issues, per capita income on the demand side is another factor contributing to regional variation [17]. Jin et al. (2020) focused on the healthcare of the elderly and analyzed the determinants of long-term care (LTC) expenditure. They revealed that a proportion of single elderly households, doctors per 100 citizens, nursing homes per 100,000 LTC benefit users, or MEO per citizen aged above 75 years would be significantly associated with LTC expenditure [18]. Population over the age of 65 was extracted as one of the key determinants of healthcare expenditure in related studies [19,20]. Healthcare expenditure disparity is also related to inequality in receiving health service (i.e., healthcare should be equal under the national insurance system but cannot be received). It is necessary to consider maintaining and improving the health system while moderating and equalizing NHE to ease medical expenses’ disparity and balance the health system performance.

Moreover, to respond flexibly to a rapidly changing and diversifying social and economic environment and to proactively implement strategic initiatives with the right people in the right places, analysis on a more diverse regional scale is necessary. However, most of these studies cover broad areas, such as state and prefecture. In Japan’s superaging society, it is thought that conducting studies in small areas and healthcare expenditure categories will provide findings that will contribute to appropriate health policies.

This study aimed to clarify the differences of the socioeconomic determinants of NHE and health system performance between subregions. The model regions are chosen among municipalities in Tokyo (central cities) and Chiba Prefecture (suburbs). MEI and MEO of NHE and inpatient (CRI) and outpatient (CRO) consultation rates were targeted. CRI and CRO were chosen to be closely related to MEI and MEO. Multiple regression analysis was performed to extract the socioeconomic determinants of regional variation. The health system performance was determined by using OECD indicators, including *health status*, *risk factors for health*, *access to care*, *quality of care*, and *health system capacity and resources*, followed by the Mann–Whitney *U* test to verify the differences between regions. 

## 2. Target Regions and Methods

### 2.1. Target Regions

Central cities and suburbs were grouped based on data from the Statistics Bureau of Japan [21]. The central cities were chosen among municipalities in Tokyo wards with populations of 500,000 or more, not classified as belonging to Tokyo special wards. The suburbs were chosen among Chiba Prefecture municipalities’ neighboring central cities, where 1.5% or more of the total residents aged 15 and older commute to work/school in the central cities (Figure 1). Chiba Prefecture’s suburbs were classified into three groups to reflect the population variance: populations of 100,000 or more, less than 100,000 and more than 30,000, and less than 30,000. Eight municipalities were randomly selected from the 100,000 or more group, 10 municipalities from the less than 100,000 and more than 30,000 group, and 9 municipalities from the less than 30,000 group [22]. Finally, 23 wards from Tokyo and 27 municipalities from Chiba Prefecture were chosen.

### 2.2. Multivariable Regression Analysis

Multiple regression analysis with a stepwise method was performed to extract factors affecting the MEI, MEO, CRI, and CRO of the central cities and suburbs. Backward elimination was conducted to choose independent variables to fit regression models by deleting some variables in Table 1.

The dependent variables are Total (MEI + MEO), MEI, MEO, CRI, and CRO. Five analyses were performed based on Equation (1), where $Y_{n}$ refers to a dependent variable (Total, MEI, MEO, CRI, or CRO), *X_m_* to an independent variable, and *b_m_* to the partial regression coefficient of each independent variable. *n* is the number of independent variables (total, MEI, MEO, CRI, CRO), and *b*_0_ is the intercept. *u* is an error term.
(1)$$Y_{n}=b_{1}X_{1}+b_{2}X_{2}+b_{3}X_{3}+\cdot \cdot \cdot +b_{0}+u$$

Ten independent variables were designed to reflect the characteristics of health service, socioeconomics, and lifestyle, including the numbers of doctors, nurses, and beds; income; the number of workers employed in primary industries; the number of entirely unemployed individuals; the percentage of population aged 65–74; the number of household members; the percentage of single households; and the percentage homeowners (Table 1). Multicollinearity was tested using variance inflation factors, and it was not detected between independent variables. Normality test of residuals was conducted using a Q–Q plot and was mainly normally distributed except for one analysis of MEI for the central cities and another of MEO for the suburbs. These two dependent variables were log-transformed in the analysis (IBM SPSS Statistics Basic V 28, IBM, New York, NY, USA).

### 2.3. Health System Performance Analysis

The health system performance analysis was performed using 18 indicators based on OECD indicators for five dimensions: *health status*, *risk factor for health*, *access to care*, *quality of care*, and *health system capacity and resources* (Table 2) [32]. For *health status*, life expectancy at birth by gender, standardized mortality ratio (SMR) by gender, and rate of certification of long-term care (LTC) or support reflecting the morbidity rate were chosen. *Risk factor* included percentages of smoking, drinking, and obesity. *Access to care* included the regional difference index of NHI and gender/age-adjusted standardized claim ratio, showing regional variation in healthcare supply. For *quality of care*, the number of home care service recipients was chosen, indicating the degree of primary care, the mortality rate of acute myocardial infarction (AMI), and the perinatal mortality rate, including childbirth trauma and complications as causes of death. *Health system capacity and resources* included medical expenses per NHI member, medical expenses per person enrolled in the healthcare system for the advanced elderly (75+), and the numbers of doctors, nurses, and hospital beds.

Health system performance was analyzed by using the relative value (*RV*) and the Mann–Whitney *U* test. *RV* is used to illustrate the characteristics of each region, and the Mann–Whitney *U* test is utilized to test the differences between regions in the central cities and suburbs. *RV* is kind of a normalized value of an indicator and enables comparison between municipalities in each region based on the indicator and characterization of the region (Equation (2)). *X* is the data of each municipality in each indicator, $\bar{x}$ is the mean value of each indicator, and σ is the standard deviation of each indicator.
(2)$$RV=\left(X-\bar{x}\right)/\sigma$$

In parallel, the Mann–Whitney *U* test was performed to identify the statistically significant differences in 18 indicators between the central cities and the suburbs (IBM SPSS Statistics Basic V 28, IBM, New York, NY, USA).

## 3. Results

### 3.1. Determinants of NHE

Descriptive statistics is shown in Table 3. Determinants of Total, MEI, and MEO of NHE and CRI and CRO of the central cities and suburbs were extracted by using stepwise multiple regression analysis (Table 4).

Total and MEI were significantly related to the number of persons per household in the central cities (0.700, 0.642, <0.001). In the suburbs, the percentage of population aged 65–74 (0.699, 0.610, <0.001), the number of hospital beds (0.354, 0.393, <0.05), and the number of workers employed in the primary industries (−0.315, −0.373, <0.05) were significantly related to Total. The adjusted R^2^ was 0.465 and 0.539, respectively.

MEO was significantly related to the number of persons per household consistent with Total and MEI in the central cities (0.760, <0.001). In the suburbs, the percentage of population aged 65–74 was significantly related (0.623, <0.001).

The percentage of population aged 65–74 (1.205, <0.001) and the percentage of unemployed (−0.726, 0.002) were significantly related to CRI in the central cities. In the suburbs, the percentage of population aged 65–74 (0.495, 0.005) and the number of hospital beds (0.398, 0.020) were significantly related to CRI. The adjusted R^2^ values were 0.620 and 0.338, respectively.

The number of household members (0.786, <0.001) and the percentage of entirely unemployed individuals (−0.360, 0.030) were significantly related to CRO in the central cities. Conversely, no significant factors were shown in the suburbs.

### 3.2. Health System Performance

The *RV* of the health system’s performance in 23 wards of the central cities revealed a close relationship with women and men’s life expectancies and drinking and smoking. The numbers of doctors, beds, and nurses were correlated, and health spending consisting of the regional healthcare expenditure index and medical expenditure by NHI for the elderly (i.e., medical care system for older senior citizens) were relatively high in all wards.

In the suburbs, the percentage of people certified for long-term care (LTC) tended to be high in areas where the SMR of women was high. In addition, the regional healthcare expenditure index and medical expenditure by NHI for the elderly were relatively low in all suburban municipalities.

The health system performance comparison between the central cities and the suburbs was evaluated using the Mann–Whitney *U* test (Table 5). In *health status*, there was no significant difference in the life expectancy and SMR of men with probabilities of 0.726 and 0.661, respectively. Conversely, women’s differences in life expectancy and SMR were significant, suggesting a relatively long life expectancy and low mortality rate in the central cities and a relatively short life expectancy and high mortality rate in the suburbs. The percentage of people certified for LTC was significantly high in the suburbs. In *risk factors for health,* all the three indicators of smoking, drinking, and obesity were significant. Smoking and drinking were relatively high in the central cities, and obesity was relatively high in the suburbs. In *access to care*, the regional healthcare expenditure index was significantly high in the central cities. The standardized claim data ratio had almost no difference (0.481). In *quality of care*, home care (care prevention) service, obstetric trauma, and AMI mortality indicators were significantly low in the central cities. In *health system capacity and resources*, the medical expenditure by NHI for the elderly and the numbers of doctors, nurses, and beds were significantly high in the central cities. In the suburbs, the medical expenditure by NHI for those other than the elderly was relatively high, and the numbers of doctors, nurses, and hospital beds were relatively low.

## 4. Discussion

### 4.1. Variance between Central Cities and Suburbs

#### 4.1.1. Different Factors Affecting NHE

In the central cities, the number of household members, the percentage of entirely unemployed, and the percentage of the population aged 65–74 were significantly related to the medical expenses (MEI, MEO) of NHE and inpatient (CRI) and outpatient (CRO) consultation rates. In the suburbs, the percentage of the population aged 65–74, the number of hospital beds, and the number of people employed in primary industries were also significantly related to the medical expenses (MEI, MEO) of NHE and inpatient (CRI) and outpatient (CRO) consultation rates (Table 4). In both regions, the percentage of the population aged 65–74 was a significant factor, suggesting that it could be one major concern to boost medical expenditure regardless of regional characteristics.

In the central cities, the number of household members was a major factor affecting healthcare expenditure. The number of household members is the average number per household in the target area. In four-fifths of the central cities, the number of members was less than two, and in all the suburbs, the number of members was over two. In the suburbs, the number of households skewed toward family households, while single-person households were prominent in the central cities. Single-person households tended to be alone more often than family households, making it easier for them to choose home treatment because even mild symptoms could be noticed late or overlooked. Furthermore, family households possibly included elderly members, who could increase medical costs, in which economic constraints might have an impact. According to governmental survey, the percentage of entirely unemployed single-person households was higher than that of households with two or more people [43]. Particularly in the central cities with convenient access to medical facilities, a clearer distinction may be made between economically insecure singles who prefer home care. Similarly, given that the percentage of completely unemployed was significantly negatively related to MEI, MEO, CRI, and CRO, economic status would be a major factor related to medical expenses and the chance to obtain health service. This suggests that economic restrictions may be a major factor in the supply and demand of medical services [44]. What is interesting is that this is more evident in the central cities. Income size, especially disposable income, is expected to affect healthcare spending, which accounts for a portion of the total consumption. After time constraints, income size was the second biggest factor restraining medical consultations, mostly driven by unemployment [45]. Another possibility would be that those insured by unions, government agencies, and other organizations shifted to the National Health Insurance because of unemployment [16].

In the suburbs, the ratio of the population aged 65 to 74 was larger than in the central cities [29,30], followed by expanding medical expenditures in the region. Medical expenditure surges as the aging progresses. Hence, a relatively poor healthcare service in the suburbs could increase NHE coupling with a high aging rate; this can be assessed by the number of beds per population, the mean length of stay, and so on. These numbers were superior in the central cities. For example, the number of beds per 100,000 population was 855 in the central cities and 559 in the suburbs; the number of doctors per 100,000 population was 548 in the central cities and 158 in the suburbs; the number of nurses per 10,000 population was 113 in the central cities and 55 in the suburbs; and the mean length of stay was 17.8 days in the central cities and 25.5 days in the suburbs in 2018 [24,25,41,42,46]. The percentage of estimated inpatients per 100,000 population was high in the suburbs, although the number of beds per capita was smaller than in the central cities. The bed occupancy rate, estimated by dividing the mean number of inpatients per day by the number of available beds, was 80 in the central cities and 79 in the suburbs, showing no significant difference. However, the mean length of stay was longer in the suburbs, and the length of stay in the suburbs caused surging medical expenses per bed, increasing the medical expenditures.

Another significant factor was the number of workers in primary industries in the suburbs, where a relatively large number of the elderly were engaged in agriculture. It was shown to be negatively related to the total medical expenditure, especially for inpatients (Table 4). Considering that the elderly account for a high proportion of medical expenditures, they decrease with the number of people engaged in agriculture, indicating that agriculture contributes to maintaining health in the suburbs, where there are many older people engaged in agriculture. Agriculture has multisocial functions, such as a place of labor, the motivation of the lifestyles of the elderly, their health promotion, and disease prevention. It retains a system of participation for the elderly, which provides employment and social participation, affects making friends, and leads to a decrease in medical expenses [47]. Since agricultural work itself is also a moderate exercise, the possibility of developing lifestyle diseases among the elderly is lower than that among nonfarmers, decreasing medical expenditures [19,48,49]. In Brazil, urban agriculture was also suggested to be a tool for health promotion due to its contribution to strengthening individual and community leadership, empowerment, and creating conducive environments for health [50]. There was also evidence of negative associations of engaging in agriculture compared with other types of employment with the prevalence of hypertension, being overweight, and obesity, and positive associations with the prevalence of being underweight and smoking [48].

The determinants of the population aged 65 and older and the number of hospital beds have been shown to contribute to regional variation [20,51,52,53]. Comparison between subdivided areas was able to extract household members and employment status according to the socioeconomic characteristics of the central cities and the suburbs, such as the entirely unemployed in the central cities and those engaged in primary industry in the suburbs, thereby identifying distinctive determinants of those areas. In the central cities where there are relatively abundant single-person households and economic constraints, access to health services is closely related to economic status and household membership, suggesting the need to promote support for single-person households and economically insecure populations. On the other hand, in the suburbs where many people are engaged in primary industries and the population of people over the age of 65 is relatively large, the number of hospital beds is the main determinant, and the length of hospital stay may be longer than in the central cities. Therefore, it was suggested that it is necessary to control hospital stay by expanding medical support and improving access to medical facilities.

#### 4.1.2. Health System Performance

The health system performance analysis clarified the characteristics and difference between the central cities and the suburbs. *Access to care*, *quality of care*, and *health system capacity and resources* were better in the central cities, while people in the suburbs had shorter life spans, higher mortality rates, and higher certification rates for long-term care (LTC) (Table 5). In the central cities, *quality of care* is considered to be superior in spite of the higher risk factors, such as smoking and drinking, due to the extensive medical system, including the numbers of hospital beds, doctors, and so on. In order to reduce the risk, a health program and/or outreach campaign to raise awareness of the risk of smoking and drinking and enhance behavior changes in everyday lives is required. On the other hand, the suburbs have concerns about life span, mortality rate, and LTC, although *risk factors of health* are lower than those in the central cities. This may result from an inadequate healthcare system, which is also shown in less *health system capacity and resources*. On the other hand, this may be closely related to elderly care. Therefore, it is necessary to improve the medical care system and the medical care program for the elderly by facilitating access, increasing subsidy, and so on. The regional characteristics of each indicator are described below.

##### Health Status

In the suburbs, the life expectancy of women was short, and their mortality rates were high. In addition, the certification rate for LTC was higher. Women tend to have longer life spans than men and have a higher rate of death from senility [21,45,54]. In addition, in Japan, the number of deaths due to childbirth has decreased since the 1940s, and women’s life spans have increased rapidly. Regarding women’s SMR, the perinatal mortality rate was considerably higher in the suburbs than in the central cities [39,40], which possibly brought the high mortality rate in the suburbs.

##### Risk Factor for Health

Smoking/drinking rates in the central cities and obesity rates in the suburbs were high. In the suburbs, many middle-aged and older people have a high proportion of obesity [55,56], said to be a cause of various lifestyle-related diseases, and obesity prevention measures are important in the surrounding areas. Furthermore, the smoking and drinking rates are high in the central cities. Although the determinants of the prevalence of smoking and drinking in the central cities in Japan are not verified, it is necessary to consider reduction measures. The National Cancer Center found that the proportion of people without cancer or cardiovascular disease for 10 years did not change with a BMI of 30 or higher (obese) or reduced to 23–27. However, the proportion increased when individuals quit smoking and reduced drinking [57]. This suggests that high rates of smoking and alcohol consumption are associated with higher rates of cancer and cardiovascular disease, leading to higher medical costs.

##### Access to Care

The regional healthcare expenditure index was high in the central cities, where the quality of care and medical resources are abundant. There was no significant difference in the standardized claim data ratio. The variation was considerable in the central cities, suggesting that access to hospitalization differs significantly, even within the central cities. In addition, the top 5 wards are concentrated in the central area of Tokyo, such as Chiyoda, Minato, Shibuya, Meguro, and Shinjuku, which have excellent transportation access; subsequently, easy access to medical care in the central cities is dependent on transportation [58,59].

##### Quality of Care

The perinatal mortality rate, the number of home care service recipients, and the AMI mortality rates are low in the central cities, and the quality of care is deemed superior to that in the suburbs. Generally, quality of care and medical expenditures are linked, which is reflected in the differences in this study’s regional healthcare expenditure index. It is not easy to achieve quality improvement and cost reduction; however, one measure is to reduce the average length of hospital stay. Increasing the number of general clinics and providing better access to medical care, although temporarily costly, lead to improved primary care, improved health status, and early detection of diseases, resulting in reduced hospital stays and costs.

##### Health System Capacity and Resources

Healthcare expenditure by the National Health Insurance for the elderly and the numbers of doctors, nurses, and beds were large in the central cities, while medical expenditure by the National Health Insurance for those other than the elderly was significant in the suburbs. Compared with the central cities, aging in the suburbs is accelerating. The LTC system for the elderly is believed to be burdened by the relatively small number of beds and poor access to medical institutions, leading to relatively low National Health Insurance medical expenditures for the elderly. The number of LTC service recipients per 100,000 population was 4.23 in the central cities and 5.29 in the suburbs [35]. In analyzing the factors affecting the medical expenditures, the number of beds was related to increased MEI and MEO in the suburbs. Therefore, it was thought that fewer doctors, beds, and nurses in the suburbs contribute to the length of hospital stay, increasing the expenditures.

### 4.2. Moderation of NHE in Central Cities and Suburbs

To moderate NHE, it is necessary to efficiently and effectively utilize healthcare resources, maintain the health of residents, and build an efficient healthcare system. Such actions aim to suppress excessive medical expenditures and maintain and improve the health system’s performance. To moderate NHE appropriately, it is necessary to consider the characteristics of each region. Given the rapid diverging regional circumstances with a hyperaging society, a more fragmented regional management would be required to optimize the healthcare service and appropriate operation. This study conducted a case study to confirm the regional variance between adjacent municipalities and suggest region-specific healthcare strategies to moderate NHE and improve the health system performance.

In the central cities, the number of household members increased medical expenditures, while the percentage of entirely unemployed individuals was negatively related to medical expenditures. The health system performance analysis showed that *risk factor for health* was higher than that in the suburbs, while *access to care*, *quality of care*, and *health system capacity and resources* were superior. This result suggests that a well-developed medical system compensates for the significant risk factors in urban areas. However, considering the relationship between the percentage of fully unemployed people and the cost of medical care, there are concerns about the quality of medical services for people with economic constraints; in turn, there are also apprehensions regarding medical care disparities and a decline in medical equality due to the widening gap between the rich and the poor caused by future increases in nonregular employment [46,55].

At present, medical expenses in the central cities are lower than those in surrounding municipalities, and it is thought that efforts should be made to distribute medical care appropriately, rather than suppressing medical expenses to optimize medical costs. For example, it would be effective to review the mitigation system of NHI premium based on income. Reviewing the target household income and the reduction ratio would reduce the burden on the poor and increase the number of people who could benefit from the enhanced health system, contributing to the appropriate distribution of healthcare. The poor here are defined as individuals eligible for the National Health Insurance premium reduction system. However, other issues arise, such as what to do about insurance premiums and the need to revise the system nationwide, where policy revisions must be considered. In addition, MHLW revealed that income disparities affect healthcare and lifestyle-related habits [48]. Lifestyles were also taken up as a *risk factor for health* in the health system performance analysis and were one of the critical indicators related to medical expenses, suggesting that it would be more important to improve lifestyles according to income in the future. In addition, it is considered that the health system revision, such as the revision of the reduction rate of the National Health Insurance premium, would not lead to a fundamental solution, and improving the economic status of the poor would be more effective.

In the suburbs, the ratio of the population aged 65 to 74 years old and the number of beds led to increased medical expenditures. In detail, the same influencing factors for Total and MEI suggest that the cost of inpatient care has a significant impact on overall medical expenditures, which should be prioritized and moderated. One of the reasons the increase in the number of hospital beds affects the expansion of medical costs is the rising cost of medical care per bed. The number of hospital beds has long been cited as a factor in increasing medical costs, but the development of home healthcare has mitigated the increase in the number of hospital beds. However, to avoid a situation where patients who should be hospitalized cannot be hospitalized, it is necessary to subdivide medical services to handle various cases. Since the utilization rate of hospital beds was comparable to that of a central city, the longer hospitalization period affects the high medical cost per number of beds, leading to an increase in inpatient medical cost. It is necessary to consider measures such as expanding medical resources to reduce the medical cost per hospital bed. It was consistent in the health system performance analysis that *access to care, quality of care, and health system capacity and resources* were lower than in the central cities, suggesting that the healthcare system might not be sufficient in the suburbs. Therefore, the priority would be to enhance the capacity and resources of the health system in the suburbs.

In addition, easy access to healthcare leads to early detection of illness and early discharge of patients; subsequently, the development of the transportation system around medical facilities would be helpful, further shortening the length of hospital stays. Japan has a “free access system” that allows patients to choose their medical institution and receive medical treatment when needed, regardless of the medical institution or department size [12]. It would be helpful to design a regional system that includes the placement of medical institutions and a transportation system that facilitates access to medical institutions, leading to the moderation of medical costs in the future.

Moreover, the number of primary industry workers decreases medical expenditures. Considering that most primary industry workers in the suburbs are elderly individuals engaged in the agriculture sector, their health promotion would be an issue. Rural communities face an evident and immediate challenge compared with urban or semiurban communities [60]. To solve this challenge in the suburbs with a high elderly population, constructing a community for the elderly would effectively support them to keep good lifestyles and improve health. Adopting a physically active lifestyle, including increasing leisure time, may contribute to a healthy aging process among the elderly [54,61,62]. Through community activity with physical movements, they could progress the motivation and vitality. If such communities centered on the elderly were designed widely in the suburbs, not only in agriculture, it could be expected to shorten the length of hospital stays and moderate the cost of inpatient care, which would moderate medical expenditures.

## 5. Conclusions

This study determined the factors affecting the regional variation of national healthcare expenditure (NHE) and subsequent health system performance between the central cities and the suburbs in Tokyo and adjacent areas in Japan. As a result, region-specific issues for moderating national healthcare expenditure and improving healthcare system performance were extracted. In the central cities, healthcare disparities need to be reduced, including a review of national health insurance, as well as health risks. In the suburbs, priority should be given to improve the healthcare of the elderly, including the expansion of health promotion programs for the elderly, and improve access to medical facilities, as well as the expansion of healthcare systems and resources.

These analyses are, however, subject to the specific limitations associated with data availability for Japan. First, data access was limited to up-to-date mortality, and sources of subregional data are not available at the municipality level for the estimates of prevalence and incidence for several diseases and their risk factors. Those subregional estimates depend on a replacement of related indicators. Second, socioeconomic parameters are limited at the individual level and in healthcare resources. For example, addressing an economic indicator by municipality will substantially improve future analysis. Finally, it targets limited regions in Japan. By adding subregions to compare, it will show the diversified region-specific properties.

Given the rapidly diversifying subregional environments, this study provided a new approach to dig insights into the variation of NHE determinants and health system performance in adjacent subregions, particularly in light of the rapid population aging process and the growing regional variations in subregions.

## Figures and Tables

**Figure 1 healthcare-10-00968-f001:**
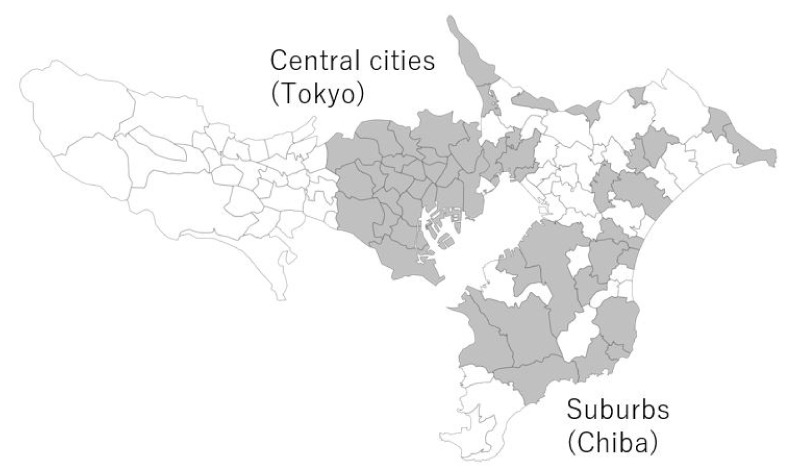
Target areas in Tokyo and Chiba Prefecture (colored in grey).

**Table 1 healthcare-10-00968-t001:** Description of independent variables.

Independent Variable		Description	Data Source
Health service	Number of doctors	Number of doctors per 100,000 population	[23]
	Number of nurses	Number of nurses per 10,000 population	[24,25]
	Number of beds	Number of beds per 100,000 population	[23]
Socioeconomic	Income	Taxable income per capita	[26]
	Number of people employed in primary industries	Number of workers employed in primary industries per 100,000 population	[27,28]
	Percentage of completely unemployed	Percentage of completely unemployed	[25,26]
	Percentage of population aged 65–74	Percentage of the population aged 65–75 and 0–74	[29,30]
Lifestyle	Number of household members	Number of persons per household	[27,29]
	Percentage of singles	Percentage of single-person households against total households	[28,31]
	Percentage of households with own houses	Percentage of households in owned houses against total households	[25,31]

**Table 2 healthcare-10-00968-t002:** OECD indicators of health system performance and their definitions.

OECD		This Study		Data Source
Indicator	Description	Indicator	Description	
*Health status*				
Life expectancy men	Years of life at birth	Life expectancy men	Years of life at birth	[33]
Life expectancy women		Life expectancy Women		
Avoidable mortality/chronic disease morbidity	Preventable and treatable deaths (per 100,000 people, age-standardized)/diabetes prevalence (% adults, age-standardized)	Standardized mortality ratio (SMR) men	A ratio between the expected number of deaths calculated and the number of deaths observed when the baseline mortality rate (number of deaths per 100,000 population) is applied to the target area	[34]
		SMR women		
Self-rated health	Population of those in poor health (% population aged 15+)	Percentage of people certified for long-term care	Population of first long-term care insurance insured persons who have been certified as requiring support or long-term care (% population aged 65+)	[35]
*Risk factors for health*				
Smoking	Daily smokers (% population aged 15+)	Smoking	Daily smokers (% population aged 20+)	[36,37]
Alcohol	Liters consumed per capita (population aged 15+), based on sales data	Alcohol	Daily drinkers (% population aged 20+)	
Overweight/obese	Population with BMI ≥ 25 kg/m^2^ (% population aged 15+)	Overweight/obese	Population with BMI ≥ 25 kg/m^2^ (% population aged 15+)	
Ambient air pollution	Deaths due to ambient particulate matter, especially PM 2.5 (per 100,000 people)	-	-	-
*Access to care*				
Population coverage, eligibility	Population covered for a core set of services (% population)	-	-	-
Population coverage, satisfaction	Population satisfied with the availability of quality healthcare (% population)	-	-	-
Financial protection	Expenditure covered by compulsory prepayment schemes (% total expenditure)	Regional healthcare expenditure index	Per capita, medical expenses in the region are indexed (1 for the whole country, age-standardized)	[13]
Service coverage	Population reporting an unmet need for healthcare (% population)	Standardized claim data ratio	Number of each medical practice that appears on the receipt is indexed (age-standardized)	[7]
*Quality of care*				
Safe primary care	Antibiotics prescribed (defined daily dose per 1,000 people)	Home care (care prevention) service	Number of home care (care prevention) service recipients (per 100,000 people)	[38]
Effective primary care	Avoidable COPD admissions (per 100,000 people, age/sex standardized)			
Effective preventive care	Mammography screening within the past 2 years (% of women aged 50–69 years)	Obstetric trauma	Perinatal mortality rate (per 1000 births)	[39,40]
Effective secondary care	30-day mortality following AMI (per 100 admissions, age-sex standardized)	AMI mortality	AMI mortality (per 100,000 people)	
*Health system capacity and resources*				
Health spending	Total health spending (per capita, USD using purchasing power parities)	Medical expenditure by National Health Insurance	Actual cost incurred at medical facilities for those other than the elderly (per capita, JPY 10,000)	[13]
			Actual cost incurred at medical facilities for the elderly (per capita, JPY 10,000)	
Doctors	Number of practicing physicians (per 1000 people)	Doctors	Number of practicing physicians (per 100,000 people)	[23,41,42]
Nurses	Number of practicing nurses (per 1000 people)	Nurses	Number of practicing nurses (per 100,000 people)	
Hospital beds	Number of hospital beds (per 1000 people)	Hospital beds	Number of hospital beds (per 100,000 people)	

**Table 3 healthcare-10-00968-t003:** Descriptive statistics of determinants.

	Obs	Mean	Std Dev	Min	Max
**Central cities**					
Medical expenses * (JPY **)	23	240,569	21,129.2	196,297	278,072
Medical expenses for inpatients (JPY)	23	100,789	12,193.8	78,476	123,425
Medical expenses for outpatients (JPY)	23	115,494	9181.7	97,350	132,347
Inpatient consultation rates (%)	23	17	1.9	14	21
Outpatient consultation rates (%)	23	774	58.1	630	859
Number of doctors	23	548	569.0	138	2411
Number of beds	23	855	774.9	201	3729
Number of nurses	23	113	106.8	40	525
Income (10,000 JPY)	23	503	191.8	336	1112
Number of people employed in primary industries (number per 100,000 employees)	23	37	38.3	0	139
Percentage of completely unemployed (%)	23	4	0.7	2	4
Percentage of population aged 65–74 (vs. population aged 0–74)	23	12	1.5	9	15
Number of household members (person)	23	2	0.2	2	2
Percentage of singles (%)	23	52	7.5	39	65
Percentage of households with own houses (%)	23	46	5.5	33	55
**Suburbs**					
Medical expenses * (JPY **)	27	276,469	23,559.6	247,285	343,974
Medical expenses for inpatients (JPY)	27	124,190	13,281.9	95,164	152,871
Medical expenses for outpatients (JPY)	27	127,829	12,711.1	112,000	167,089
Inpatient consultation rates (%)	27	22	2.4	17	26
Outpatient consultation rates (%)	27	804	65.4	706	967
Number of doctors	27	158	262.1	24	1476
Number of beds	27	559	646.8	0	3021
Number of nurses	27	55	60.2	7	348
Income (10,000 JPY)	27	300	37.7	252	382
Number of people employed in primary industries (number per 100,000 employees)	27	1515	1563.4	4	6421
Percentage of completely unemployed (%)	27	4	0.7	3	6
Percentage of population aged 65–74 (vs. population aged 0–74)	27	18	3.3	13	30
Number of household members (person)	27	3	0.2	2	3
Percentage of singles (%)	27	28	6.4	18	45
Percentage of households with own houses (%)	27	78	11.4	50	96

* Medical expenses include those of inpatients, outpatients, and dental treatments. ** approximately JPY 1 to USD 0.01.

**Table 4 healthcare-10-00968-t004:** Multiple regression analysis results.

Dependent Variables	Independent Variables	Central City		Suburbs	
		Coefficients	*p*	Coefficients	*p*
Medical expenditure for medical treatment (Total)	Persons per household	0.700	<0.001	-	-
	Percentage of population aged 65–74	-	-	0.699	<0.001
	Hospital beds	-	-	0.354	0.017
	Workers employed in primary industries	-	-	−0.315	0.042
	Adjusted R^2^	0.465		0.539	
Medical expenditure of inpatient (MEI)	Persons per household	0.642	<0.001	-	-
	Percentage of population aged 65–74	-	-	0.610	<0.001
	Hospital beds	-	-	0.393	0.012
	Workers employed in primary industries	-	-	−0.373	0.023
	adjusted R^2^	0.384		0.556	
Medical expenditure of outpatient (MEO)	Persons per household	0.760	<0.001	-	-
	Percentage of population aged 65–74			0.623	<0.001
	Adjusted R^2^	0.558		0.364	
Consultation rate of inpatient (CRI)	Percentage of population aged 65–74	1.205	<0.001	0.495	0.005
	Percentage of completely unemployed	−0.726	0.002	-	-
	Hospital beds	-	-	0.398	0.020
	Adjusted R^2^	0.620		0.338	
Consultation rate of outpatient (CRO)	Persons per household	0.786	<0.001	-	-
	Percentage of completely unemployed	−0.360	0.030	-	-
	Adjusted R^2^	0.529		-	

**Table 5 healthcare-10-00968-t005:** Mann–Whitney *U* test of the health system performance between the central cities and the suburbs.

Dimension	Indicator	Frequency	Mean Rank	Rank Sum	Mann–Whitney *U*	*Z*	Asymptotic Significance (2-Side)
*Health status*	Life expectancy men	23	26.28	604.5	292.5	−0.351	0.726
		27	24.83	670.5			
	Life expectancy women	23	30.93	711.5	185.5	−2.440	0.015
		27	20.87	563.5			
	SMR men	23	24.52	564.0	288.0	−0.438	0.661
		27	26.33	711.0			
	SMR women	23	19.35	445.0	169.0	−2.755	0.006
		27	30.74	830.0			
	Percentage of people certified for long-term care (LTC)	23	18.22	419.0	143.0	−3.261	0.001
		27	31.70	856.0			
*Risk factor for health*	Smoking	23	33.24	764.5	132.5	−3.466	<0.001
		27	18.91	510.5			
	Drinking	23	35.22	810.0	64.0	−4.710	<0.001
		26	15.96	415.0			
	Overweight/obese	23	19.00	437.0	161.0	−2.911	0.004
		27	31.04	838.0			
*Access to care*	Indices of regional healthcare expenditure	23	38.00	897.0	0.0	−6.045	<0.001
		27	14.00	378.0			
	Standardized claim data ratio	23	17.78	409.0	97.0	−0.705	0.481
		10	15.20	152.0			
*Quality of care*	Home care (care prevention) service	23	13.93	320.5	44.5	−3.252	0.001
		27	25.79	309.5			
	Obstetric trauma	22	13.22	419.0	73.5	−4.614	<0.001
		13	31.70	856.0			
	AMI mortality	23	15.20	349.5	143.0	−3.261	0.001
		27	34.28	925.5			
*Health system capacity and resources*	Medical expenditure by National Health Insurance for those other than the elderly	23	15.43	355.0	79.0	−4.506	<0.001
		27	34.07	920.0			
	Medical expenditure by National Health Insurance for the elderly	23	39.00	897.0	0.0	−6.044	<0.001
		27	14.00	378.0			
	Doctors	23	37.48	862.0	35,000	−5.363	<0.001
		27	15.30	413.0			
	Nurses	23	33.52	771.0	126,000	−3.591	<0.001
		27	18.67	504.0			
	Hospital beds	23	30.74	707.0	190,000	−2.346	0.019
		27	21.04	568.0			

## Data Availability

https://doi.org/10.5061/dryad.h18931znw (accessed on 1 May 2021).

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
