# Peer review of "Regional Variation in National Healthcare Expenditure and Health System Performance in Central Cities and Suburbs in Japan"

_healthcare, 2022, doi:10.3390/healthcare10060968_

Round 1

Reviewer 1 Report

Overall, the layout of the paper is correct. It is well-written, in terms of formal requirements, and, in addition to the theoretical background, it also contains a quantitative analysis conducted using specific statisctical methods, which should be seen as its strength. The author(s) correctly conducted the scientific discussion based on the appropriate research methods. The weakest point of the paper is its final part, which requires radical substantive corrections. In the paper the following should be corrected: (1) currently, the introduction does not precisely define the research gap based on a literature query conducted on the basis of internationally recognized and prestigious databases such as Scopus or Web of Science Core Collection. The introduction lacks a clearly stated purpose, i.e., the paper aims at...; (2) in terms of the literature review, the text of the paper lacks a clearly defined key that would allow for a correct literature search. What keywords were chosen to make the right choice of literature? Was the selection made on the basis of abstracts or an analysis of the content of entire articles? These questions should be answered in the text of the publication. (3) in general, the results of analyses and theoretical considerations do not raise serious objections. Tables and figure do not seem to be difficult to interpret. In the central part of the article, a bit more in-depth analysis and interpretation of the results should be made. In addition, the reader should be indicated what are the differences and similarities between the results of the author's (authors') calculations and the existing research results, i.e., those published so far in the scientific literature. In particular, it is necessary to answer the question asked in the paper what the implications of the results for researchers are?; (4) the conclusion is a very weak part of the paper. The final part of the paper should contain basic conclusions drawn from the considerations presented earlier. In particular, the conclusions should thoroughly and clearly answer the question about the contribution of the research results presented in the paper to science. Next, it is necessary to underline what the implications of the presented results for the practice are, and what are the limitations for the conducted research. In addition, it should be determined whether the results are unique in a certain context and what further research possibilities are available in this field in the future; (5) the article should be proofread by a certified English translator. Overall, the review is positive. The paper may be published in this journal on condition that the remarks given in the review are taken into consideration and the approval of the editor-in-chief is obtained.

Author Response

Thank you for your insightful and generous feedbacks to the manuscript. I carefully revised as following your comments and suggestion. I believe to meet your suggestions through the manuscript. Revisions are in blue highlight.

  • The introduction does not precisely define the research gap based on a literature query conducted on the basis of internationally recognized and prestigious databases such as Scopus or Web of Science Core Collection. The introduction lacks a clearly stated purpose, i.e., the paper aims at...;

Response: Literatures are modified to those from Scopus, Web of Science, etc. The purpose is also revised. Briefly, it focused on the comparison in adjacent subdivided regions. It was assumed that with the rapid diversification of regions, a more segmented regional survey would find it more appropriate to reduce NHE and improve health system performance in the context of micro regional management. However, previous studies targeting regional differences were mostly targeted state or prefectural level. Therefore, this study literally aimed to clarify the difference of NHE determinants and health system performance between subdivided regions and explore the suggestion to moderate NHE and improve Health system performance.

  • Literatures: p.4, line 54-85
  • Clarifying Purpose: p.2, line 13-15, 36-38; p.3, line 86-90

  • in terms of the literature review, the text of the paper lacks a clearly defined key that would allow for a correct literature search. What keywords were chosen to make the right choice of literature? Was the selection made on the basis of abstracts or an analysis of the content of entire articles? These questions should be answered in the text of the publication.

Response: Keyword were chosen on the basis of abstract including analyses of the content. As your suggestion, they were modified to make the content clearer. modified the words that describe the National Health Expenditure (NHE) of each country so that they can be searched. Keywords were modified, for example, “regional differences” “regional disparity” to “regional variation” to provide consistency, “National health care expenditure (NHCE)” to “National healthcare expenditure (NHE)” to allow for a correct literature search.

  • 2, Title: Regional variation, National healthcare expenditure
  • 2, Keywords: Japan was added.

  • in general, the results of analyses and theoretical considerations do not raise serious objections. Tables and figure do not seem to be difficult to interpret. In the central part of the article, a bit more in-depth analysis and interpretation of the results should be made. In addition, the reader should be indicated what are the differences and similarities between the results of the author's (authors') calculations and the existing research results, i.e., those published so far in the scientific literature. In particular, it is necessary to answer the question asked in the paper what the implications of the results for researchers are?;

Response: Please understand that Tables are descriptive because they literally describe the current situation in Japan, followed by referred to develop the suggestion for healthcare strategies and policies written in Discussion. I strongly agree that a more in-depth analysis and interpretation of the Discussion should be made including the comparison with previous research and implications of the study and revised them (p.10, line 279-292). Regarding the implications of the results, this study insists the importance of the regional analysis in subdivided regions and newly findings based on it were highlighted (p.9, line 230-232, p.11, line 354-358).  Briefly, household member and economic constraints are significant specific in central cities, hence, healthcare resource (supply sector) are in the suburbs. Based on these, I suggested regional specific NHE moderation strategy examples. Since there have been few comparative studies on health system performance in local governments, 4.1.2 mostly present the current situation in Japan.

  • the conclusion is a very weak part of the paper. The final part of the paper should contain basic conclusions drawn from the considerations presented earlier. In particular, the conclusions should thoroughly and clearly answer the question about the contribution of the research results presented in the paper to science. Next, it is necessary to underline what the implications of the presented results for the practice are, and what are the limitations for the conducted research. In addition, it should be determined whether the results are unique in a certain context and what further research possibilities are available in this field in the future;

Response: The conclusion is revised to describe the contribution of this study and underline the implication for the practice and the limitation (p. 12. line 438-500)

  • the article should be proofread by a certified English translator.

Response: The manuscript will be edited at the publication.

Reviewer 2 Report

The topic is important and interesting. However, there are several problems throughout this manuscript.

  1. The literature review is quite weak. Although the topic is interesting, how do we know that this topic has not been done before. What is the research gap between this study to the literature?
  2. Equation 1 is not satisfactorily explained. What doest n stands? What is bo?
  3. Equation 2 is also not well explained. I try to understand this equation and look at Table 4, but it seems not consistent with the equation. I may miss something here, but the methodology part needs to revise seriously to make the reader can follow your work.
  4. In the abstract, the author mentions the stepwise regression, but it seems that there is only multiple regression provided in the methodology section.
  5. In Table 3, there are four columns of results. The author should add some description to indicate which column is the coefficient and which is the p-value?
  6. In the result section, I suggest providing the data description.
  7. The conclusion should focus on the main findings, and the recommendation and limitations of the study must be added.

 Overall, the paper is not quite well written, and many unclear parts should be clarified to the reader.

Author Response

Thank you for your insightful and generous feedbacks to the manuscript. I carefully revised as following your comments and suggestion. I believe to meet your suggestions through the manuscript. Revisions are in blue highlight.

  1. The literature review is quite weak. Although the topic is interesting, how do we know that this topic has not been done before. What is the research gap between this study to the literature?

Response: Literature review is modified including research gap (p.4, line 54-90). Briefly, it focused on the comparison in adjacent subdivided regions. It was assumed that with the rapid diversification of regions, a more segmented regional survey would find it more appropriate to reduce NHE and improve health system performance in the context of micro regional management. However, previous studies targeting regional differences were mostly targeted state or prefectural level. Therefore, this study aimed to clarify the difference of NHE determinants and health system performance between subdivided regions and explore the suggestion to moderate NHE and improve Health system performance.

  1. Equation 1 is not satisfactorily explained. What doest nstands? What is bo?

Response: Explanation is added in p.3. line 122-123. n is the number of independent variables (total, MEI, MEO, CRI, CRO) and b0 is the intercept.

  1. Equation 2 is also not well explained. I try to understand this equation and look at Table 4, but it seems not consistent with the equation. I may miss something here, but the methodology part needs to revise seriously to make the reader can follow your work.

Response: The health system performance was analyzed by using the relative value (RV) and the Mann-Whitney U test in parallel. RV was estimated for each municipality in 18 indicators to understand the differences in the health system performance between central cities and the suburbs through comparison of each indicator between municipalities, which was calculated using Equation (2). To show that the difference between the two regions is significant, Mann-Whitney U test was performed by using 18 indicators of both regions (p.6, line 151 - 159).

  1. In the abstract, the author mentions the stepwise regression, but it seems that there is only multiple regression provided in the methodology section.

Response: Backward elimination is used to the stepwise multiple regression analysis. Description of stepwise multiple regression analysis is added in the methodology section (p.3, line 116-118).

  1. In Table 3, there are four columns of results. The author should add some description to indicate which column is the coefficient and which is the p-value?

Response: Column description is added in Table 3.

  1. In the result section, I suggest providing the data description.

Response: Table 1 and Table 2 show the data descriptions of the variables. I'm afraid of duplication.

  1. The conclusion should focus on the main findings, and the recommendation and limitations of the study must be added.

Response: The conclusion is revised to clearly describe the contribution of this study and underline the implication for the practice and the limitation (p. 12. line 438-459).

Round 2

Reviewer 2 Report

The authors make a lot of changes to improve the paper, however there are some comments to improve the quality of the paper

  1. Eq1 is still not complete, error term is missing.
  2. I mean adding the summary statistics table for  describing the data.
  3. the explaination of Mann-whitney U does not complete. How it links to RV?. This is very important process as it will help the reader to follow Table 4.
  4. Table 3, p-value should be presented in same notation.
  5. line 118 is not complete.  some...?
  6. It is better to refer the table results in discussion part. At this presentation, it quite difficult to follow.
  7. In this work, many factors are investiagted as the factor of NHE, but the topic looks like the author want to investigate the reginal variation in NHE. Please consider to make your topic corresponds to your works            Although, the author try a hard work to revise the manuscript. It still has a lot of concerns. please carefully improved your work.

Author Response

Thank you for your generous comments to the manuscript. I carefully revised as following your comments and suggestion. I believe to meet your suggestions through the manuscript. Revisions are in blue highlight.

  1. Eq1 is still not complete, error term is missing.

Response: Error term is added (line 122).

  1. I mean adding the summary statistics table for describing the data.

Response: The summary statistics table was added (Table 3).

  1. the explaination of Mann-whitney U does not complete. How it links to RV?. This is very important process as it will help the reader to follow Table 4.

Response: Methods were amended to clearly describe RV and Mann-Whitney U test. Both were conducted for different purpose. RV is to illustrate the characteristics of each region and the Mann-Whitney U test is to test the difference between regions of central cities and the suburbs. RV is kind of normalized value of an indicator and enables comparison between municipalities in each region based on the indicator, and to characterize the region. On the other hand, Mann-Whitney U test was performed to test a difference between two regions according to indicators (p.6, line 150- 159). RV results is described in Results (line 187-195). Supplement data upload is not available at present. If necessary, it will be attached.

  1. Table 3, p-value should be presented in same notation.

Response: p is amended in same notation (Table 4).

  1. line 118 is not complete.  some...?

Response: “variables” was added to the sentence (line 117).

  1. It is better to refer the table results in discussion part. At this presentation, it quite difficult to follow.

Response: Tables are referred in discussion part (line 216-224, line 297-300).

  1. In this work, many factors are investigated as the factor of NHE, but the topic looks like the author want to investigate the reginal variation in NHE. Please consider to make your topic corresponds to your works.

Response: This study aimed to clarify the differences of the socioeconomic determinants of NHE, and health system performance between subregions (p.2, line 89-90). Please note that the factors of NHE were used to compare the regional variations in adjacent regions.
